# A Refined Method for Studying Foraging Behaviour and Body Mass in Group-Housed European Starlings

**DOI:** 10.3390/ani12091159

**Published:** 2022-04-29

**Authors:** Melissa Bateson, Ryan Nolan

**Affiliations:** Biosciences Institute, Newcastle University, Newcastle upon Tyne NE2 4HH, UK; ryan.nolan@liverpool.ac.uk

**Keywords:** passerine bird, foraging, fat, body mass, mass gain, energy requirement, adaptive regulation

## Abstract

**Simple Summary:**

Small birds such as European starlings respond rapidly to environmental challenges by losing or gaining weight. Laboratory studies of these birds are therefore useful for understanding how the environment affects body weight. However, practical constraints including the need to catch birds frequently for weighing has meant that birds are often housed alone in small cages for such studies. Such conditions are unnatural and are likely to cause stress. Consequently, the data obtained from these studies are unrepresentative of wild birds. Here, we describe a novel technology based on smart feeders that permits continuous recording of foraging behaviour and body masses from starlings housed in groups in large indoor aviaries that permit more natural behaviour. We show that the birds quickly learn to use the feeders and that the system delivers detailed real-time data on foraging behaviour and body mass, without the need for frequent catching. The data obtained allowed us to study how the foraging decisions that a bird makes within a single day affect its body weight that day. These improvements in the quality of the data that we are able to collect will help inform our understanding of the environmental causes of weight gain and obesity.

**Abstract:**

Laboratory experiments on passerine birds have been important for testing hypotheses regarding the effects of environmental variables on the adaptive regulation of body mass. However, previous work in this area has suffered from poor ecological validity and animal welfare due to the requirement to house birds individually in small cages to facilitate behavioural measurement and frequent catching for weighing. Here, we describe the social foraging system, a novel technology that permits continuous collection of individual-level data on operant foraging behaviour and body mass from group-housed European starlings (*Sturnus vulgaris*). We report on the rapid acquisition of operant key pecking, followed by foraging and body mass data from two groups of six birds maintained on a fixed-ratio operant schedule under closed economy for 11 consecutive days. Birds gained 6.0 ± 1.2 g (mean ± sd) between dawn and dusk each day and lost an equal amount overnight. Individual daily mass gain trajectories were non-linear, with the rate of gain decelerating between dawn and dusk. Within-bird variation in daily foraging effort (key pecks) positively predicted within-bird variation in dusk mass. However, between-bird variation in mean foraging effort was uncorrelated with between-bird variation in mean mass, potentially indicative of individual differences in daily energy requirements. We conclude that the social foraging system delivers refined data collection and offers potential for improving our understanding of mass regulation in starlings and other species.

## 1. Introduction

Wild bird species such as the European starling (*Sturnus vulgaris*) provide valuable animal models for studying the biology of foraging behaviour and the regulation of fat reserves and body mass [1,2]. A well-developed theoretical literature in evolutionary ecology argues that birds should adaptively regulate their fat reserves to optimize the trade-off between the increased starvation risk caused by having too little fat and the increased predation risk caused by having too much [3,4,5,6,7]. These optimality models predict how birds should strategically regulate their body mass and the shape of their mass gain trajectory over the day in response to environmental challenges including perceived predation risk and unpredictable access to food. Laboratory experiments on birds have been central in testing these models of adaptive mass regulation. For example, numerous experimental studies on starlings have confirmed the theoretical prediction that limited and unpredictable access to food should cause birds to gain mass as insurance against starvation [2,8,9,10,11]. This ‘insurance hypothesis’ has recently been suggested as a possible evolutionary explanation for the increased odds of overweight observed in food insecure humans, suggesting that studies of birds could provide new insights into the environmental causes of the human obesity epidemic [12,13,14,15].

Despite the above achievements, progress in understanding the behavioural and physiological mechanisms underlying body mass regulation in birds has been limited by the practical difficulties of obtaining high quality individual-level data on body mass, foraging behaviour, and physiology from birds housed in ecologically valid conditions in the laboratory. Moreover, the common practice of individually housing birds in small cages that facilitate behavioural measurements and frequent capture for manual weighing is likely to be chronically stressful for social species such as the starling. We therefore set out to demonstrate that high quality individual-level data on foraging behaviour and body mass, which are necessary for testing mechanistic hypotheses, can be obtained from group-housed starlings living in large aviaries that simultaneously provide greater ecological validity and improved animal welfare compared with previous methods.

Laboratory studies in birds and mammals have identified a number of methodological variables that are likely to affect the ecological validity of studies of foraging behaviour and mass regulation. The first of these is cage size, which is potentially important because it affects the distances that animals can walk, run, or fly and hence their potential energy expenditure through physical activity. While it is generally assumed that increased body mass under conditions of unpredictable food results from increased energy intake [16,17,18], reduced energy expenditure is also likely to be important; studies of both starlings and zebra finches (*Taeniopygia guttata*) showed that birds respond to unpredictable food by reducing their physical activity [2,19]. Cage furnishing also affects the range of behaviour patterns that animals can perform in the laboratory and hence their energy expenditure. In rats, the offspring of mothers that are undernourished during pregnancy become obese when housed in standard laboratory cages without the opportunity for exercise, but when provided with daily access to a running wheel, they engaged in voluntary exercise and maintained normal body mass [20,21]. Thus, housing animals in small barren cages might constrain them to be sedentary and remove a potentially important mechanism that is present in the natural environment for regulating energy balance.

Housing social animals in groups is likely to be important for body mass for two reasons. First, group housing permits social behaviour, which can form an important contribution to an individual’s time budget and contribute to energy expenditure. In songbirds such as the starling, song is primarily used in intra-specific communication and is energetically costly [22]. Second, group housing can affect body mass by increasing competition for food. Dominant individuals in a group can displace subordinate animals from a monopolizable food source, creating inequalities within the group in the predictability of food access. In starlings, as predicted by the insurance hypothesis, the effects of unpredictable food access on body mass are greater when there is more competition between birds for access to feeders [2]. Furthermore, experimentally increasing a subordinate bird’s rank in a group (by removing the most dominant individual) causes a decrease in its body mass [23]. Thus, housing animals individually removes opportunities for energetically expensive social behaviour, and also removes competition for food, reducing unpredictability in food and thereby eliminating an important source of individual variability in body mass that is present in the natural environment.

All experimental studies of foraging behaviour can be classified as either open or closed-economy designs [24]. In an open economy experiment, a subject is placed in the experimental manipulation for part of the day and its total daily food intake is determined by the amount of food provided by the experimenter outside of this period. For example, in many psychological studies, subjects are maintained at a fixed percentage of their free-feeding mass, regardless of the amount of food eaten in the experimental session. In contrast, in a closed economy experiment, the manipulation is typically in place for the entire day, and the subject’s response to the manipulation completely determines its daily intake. Theoretical models show that the optimal proportion of time a subject should spend foraging depends on the rate at which it can obtain food, but that the direction of this relationship depends on whether the subject is in open or closed economy [25]. Thus, the type of economy potentially influences the effects that will be observed in experiments. Closed economy experiments provide a better simulation of the situation present in natural environments.

In summary, to perform ecologically valid studies of body mass regulation and understand the behavioural, psychological, and physiological mechanisms underlying this, we need experimental laboratory studies that incorporate the following features. First, subjects must be housed in enclosures that are large enough and contain the furnishings necessary to allow performance of a natural range of behaviour. Second, subjects of social species should be housed in groups that permit social interactions and the formation of dominance hierarchies. Third, studies must be performed in closed economy, whereby the subjects’ adjustments to the experimental conditions they are placed under directly determine their daily food intake. Finally, human intervention, especially catching and restraint, which are stressful and likely to be perceived by subjects as predation attempts, should be limited as far as possible.

Here, we describe the social foraging system (SFS), a novel technology that allows remotely operated experimental manipulation of food access and permits continuous collection of individual-level data on operant foraging behaviour. We present observational data on the patterns of foraging behaviour and mass gain that can be recorded using the SFS as well as the methods that we have developed to train the birds.

The SFS is based around smart feeding stations that identify individual birds with radio frequency identification (RFID), weigh them with an integrated electronic balance, record foraging behaviour using pecks on an illuminable pecking key, and control food access using a retractable food hopper. The feeding stations are connected to a computer that controls the operant schedule in place on the pecking key and access to the food hopper and collects data on bird identities, body masses, and foraging behaviour continuously between dawn and dusk each day (starlings do not feed at night).

A potential challenge raised by the SFS is the training of the birds. In the past, we have conducted operant training in individual cages where individual experience can be controlled and progression based on individual performance criteria (e.g., [26]). Starlings are social foragers and we speculated that group training might increase the willingness of more neophobic individuals to approach the SFS and feed from the hopper when they saw bolder birds foraging there. However, we were also concerned that during operant training, some birds might learn to scrounge food rewards earned by other individuals, rather than learning to peck the keys for food themselves. Over a series of pilot studies not described here, we developed a protocol for training starlings to forage from the SFS in groups of three birds. Six of the birds used in the current study were involved in these earlier trials and were trained to forage from the SFS prior to the start of the current study. However, the other six birds were experimentally naïve and we used this study as an opportunity to test a new social training protocol for the SFS.

We start by describing the acquisition of operant key pecking in naïve group-housed birds. Our aim was to discover whether starlings can learn key pecking without the need for social isolation. Following successful operant training, we report the observational foraging and body mass data from two groups of six birds maintained on a fixed-ratio operant schedule under closed economy for 11 consecutive days. Our aims were to describe the patterns of foraging behaviour and mass gain within the day within birds; to establish the repeatability of these measurements between days; and, to explore how variation in foraging behaviour is related to variation in body mass both within and between birds. We discuss the strengths and limitations of the SFS, focusing on both the quality of the scientific data obtained and the likely welfare of the birds.

## 2. Materials and Methods

### 2.1. Ethical Statement

The study adhered to ASAB/ABS guidelines for the use of animals in research. Birds were taken from the wild under Natural England permit 20121066 and the research was completed under UK Home Office licence PPL P038AB1D3 with the approval of the Animal Welfare and Ethical Review Body at Newcastle University. Reporting followed the ARRIVE 2.0 guidelines [27].

### 2.2. Subjects and Basic Husbandry

Subjects were 12 adult European starlings (*Sturnus vulgaris*), six males and six females, originally caught from the wild in March 2019 at Seaton Sluice, Northumberland, UK. Birds were sexed using bill and iris colour [28]. Tarsus length was measured twice for the right and left leg using digital callipers and a mean tarsus length calculated for each bird as a measure of skeletal size [29].

Prior to the study described in the current paper, which took place in March 2020, the birds were group-housed in a single indoor ‘home’ aviary (280 cm wide × 300 cm deep × 255 cm high). The aviary was without windows and was artificially lit and ventilated. The temperature and humidity were maintained at ~18–20 °C and ~40% respectively. The lights were on between 0800–1715, providing 9.25 h daylight. Between 0800 and 0815, the lights gradually increased in intensity to simulate dawn and between 1700–1715, the lights dimmed gradually to simulate dusk and allow birds to settle for the night. The aviary was furnished with rope perches, wood chippings on the floor, a water bath, and a drinker providing ad libitum clean water supplemented with vitamins. Birds were fed ad libitum on commercial poultry starter crumb (Special Diets Services Poultry Starter; henceforth ‘crumb’)—a homogeneous complete diet for starlings—supplemented with softbill mix (Orlux Universal Softbill Food), fresh fruit, and live mealworms.

For the study described in the current paper, the birds were caught and transferred to experimental aviaries described below. After completion of the data collection, the birds were retained in the experimental aviaries for an experimental study not reported here. In April 2020, following completion of the latter experiment and inspection by a veterinary surgeon, the birds were released to the wild at the site of original capture.

### 2.3. Experimental Aviaries and the Social Foraging System

Experimental aviaries were identical to the home aviary described above with the exception that each aviary was equipped with one SFS station for every three birds present in the aviary. The SFS was built to our specifications by Campden Instruments, Loughborough, UK. A single SFS station (Figure 1) comprised a motorized retractable food hopper filled with crumb and an illuminable pecking key, both of which could only be accessed via a wooden perch designed to accommodate a single bird. The perch was mounted at the apex of a smooth plastic pyramid designed to prevent other birds from perching on the station and protect the balance and RFID aerial that were located beneath it. The pyramid was mounted on an electronic balance that measured to a resolution of 0.01 g. The RFID aerial was tuned to read microchips that were glued to plastic leg rings worn by a starling standing on the perch. The SFS stations were connected to a single computer in an adjacent room running Whisker experimental control software [30], a custom-written programme (Starfeeder) that managed the RFID and mass data, and additional custom-written Whisker ‘client’ programmes specific to different phases of the study. The computer controlled the operant schedule in place on each station and collected continuous data on the identity and masses of birds visiting each station, plus any key pecks and food rewards delivered. The data files written by the Whisker client programmes could be accessed in real-time by custom-written R scripts [31] that produced summaries of the birds’ masses and key pecking behaviour for the current day. These latter summaries were checked a minimum of three times daily and were used for welfare monitoring purposes.

Husbandry in the experimental aviaries took place between 1600 and 1700 daily throughout the study. Birds were fed on crumb that was available either from the SFS hopper or from ad libitum bowls that were provided following operant training sessions. The diet was supplemented with four live mealworms per bird given during daily husbandry and supplied in spatially separated bowls to prevent one bird from monopolizing the worms.

### 2.4. Welfare Monitoring

A welfare monitoring protocol was designed that takes advantage of the mass data provided by the SFS and hence avoids unnecessary catching of the birds (Appendix A). Catching for manual weighing was only required by the protocol in the period of habituation, before birds started perching on the SFS and recording automatic body masses.

### 2.5. Study Design

The study was observational with no experimental manipulation. Data on foraging behaviour and body mass are described from two groups of six birds housed in separate experimental aviaries and maintained under closed economy for 11 consecutive days. The design is summarized in Figure 2.

### 2.6. Operant Training

Three days prior to being transferred to the experimental aviaries for the start of operant training, the diet of the birds in the home aviary was restricted to ad libitum crumb in order to habituate them to the diet that would be available from the SFS. At ~1200 on the third day, 12 birds were caught from the home aviary, manually weighed, and fitted with two coloured plastic leg rings, each of which had a unique microchip attached. Birds wore two microchips to guard against identification failure in the event that one microchip fell off, broke, or was not read due to poor alignment with the aerial in the SFS. Six of the birds had been trained to forage from the SFS ~6 months previously and were released directly into a single experimental aviary equipped with two SFS stations running the continuous foraging operant schedule (see below). The other six birds were experimentally-naïve and required operant training.

Training was conducted in groups of three to facilitate monitoring of individual performance during the acquisition of key pecking. Three birds were released into two experimental aviaries (three males in one and three females in the other), each equipped with a single SFS station. The first phase of operant training was to habituate the birds to the SFS stations and to feeding from the SFS hopper. Throughout this phase, crumb was provided ad libitum by permanently raising the hoppers of the SFS stations between 0800–1645. Birds were initially encouraged to approach the SFS stations by placing two bowls of crumb and two bowls of mealworms (highly attractive to starlings) on the base of each station. Feeding was monitored by watching the birds via a live CCTV camera and weighing the bowls and SFS hopper at the end of each day. When the birds started feeding from the bowls, they were removed, and dried mealworms were manually placed in the SFS food hopper entrance to attract the birds to start feeding from the hopper. When the birds started feeding from the hopper, the dried mealworms were discontinued, and the birds were henceforth restricted to feeding on the crumb in the hopper. Throughout habituation, the number of stable masses recorded per bird each day was used to monitor the foraging effort and body mass of each bird and the daily decrease in the mass of the food hopper to monitor crumb consumption by each trio of birds. Until individual birds started perching on the balance, they were caught and manually weighed every second day. As soon as all the birds in an aviary were perching on the SFS and feeding from the hopper, the aviary progressed to the next phase of training.

The second phase of operant training was to teach the birds to peck the lit key on the SFS to raise the hopper and hence obtain access to food. During this phase, a daily operant training session ran from 0830 until ~1400 and was followed by the provision of bowls of ad libitum crumb placed on the base of the SFS between the end of the session and 1645 when birds were food deprived until the start of the session the following morning. The SFS hoppers remained in the inaccessible retracted position outside of the training sessions.

Operant training began with daily sessions of auto-shaping, whereby illumination of the pecking key on the SFS for 15 s predicted unconditional raising of the food hopper for 15 s, starting when the key light extinguished, followed by a 200-s inter-trial interval (ITI) during which the hopper was retracted and the key light was unlit. The aim of this schedule was to set up a Pavlovian association between the lit key and food reward. The acquisition of this association typically results in a conditioned response, whereby the birds spontaneously start to direct appetitive pecks at the lit key (auto-shaping). A peck to the lit key was reinforced by immediate hopper raising, thus additionally establishing an instrumental association between pecking the lit key, and immediate food reward. Daily sessions terminated after 90 trials or 6 h. This unconditional auto-shaping training continued daily until all the birds in an aviary had started pecking the lit key—each bird was required to make a minimum of three pecks in a session before the entire aviary could progress to the next phase of training. When this criterion was met, hopper raising was made conditional on a bird pecking the illuminated key. The stimulus time was maintained at 15 s, but the feeding time was reduced to 5 s and the ITI to 66 s. The shorter feeding time was designed to reduce the potential for birds to scrounge food produced by others, which might impair them from learning the association between pecking and food. The session terminated after 270 trials or 6 h. This conditional training continued daily until all the birds in an aviary were pecking the lit key a minimum of three times in a session. When this criterion was met, the aviary progressed to continuous foraging (see below) and as soon as both aviaries of naïve birds had reached this point, the six trained birds were united in a single experimental aviary equipped with two SFS stations running the continuous foraging schedule.

### 2.7. Continuous Foraging under Closed Economy

Following operant training, the birds were maintained in a closed economy, obtaining all of their daily food from the SFS (with the exception of the four mealworms given during husbandry). The default position of the food hopper was lowered so that food was unavailable and the key light on the SFS was illuminated to indicate that the SFS station was available for foraging. Food access was delivered on a ratio schedule, whereby a single peck at the lit key caused the key light to extinguish and the hopper to raise for 5 s (henceforth referred to as ‘a reinforcement’). At the end of the food access period, the hopper lowered and there was a 2-s ITI before the key re-lit, signalling the start of the next available trial. The two SFS stations in each aviary operated independently from one another, meaning that it was possible for two starlings in an aviary to forage simultaneously on different stations. The operant sessions began at dawn each day (0800) and ended at lights-off (1715). Thus, the maximum number of reinforcements available per day from a single SFS station was 4162 (calculated assuming that a bird always pecked the lit key one second after it illuminated). Continuous foraging continued for 11 consecutive days in each of the two experimental aviaries.

### 2.8. Outcome Variables

#### 2.8.1. Operant Foraging Behaviour

In the current study, a single peck to a lit key always resulted in the food hopper raising for 5 s, meaning that the number of pecks equated to the number of reinforcements earned. The time of each peck was recorded, and the peck was attributed to the bird currently on the perch. Therefore, number of reinforcements earned was available at the individual bird level. Reinforcements earned was expressed as the rate × h^−1^.

#### 2.8.2. Food Consumption

Total crumb consumption in each aviary was estimated daily by calculating the difference in the mass of the SFS food hoppers between the beginning and end of the day and subtracting any crumb collected in a spill tray located beneath each hopper. Therefore, consumption data were only available at the aviary level, but were expressed as g × bird^−1^·day^−1^ for ease of comparison between groups of different sizes.

#### 2.8.3. Body Mass

In all experiments, body masses were recorded each day between lights-on and lights-off; each mass was assigned to the microchip recorded from the bird currently on the perch. Therefore, body mass was available at the individual bird level. The balances measured masses at a frequency of 6 Hz. A stable mass was recorded for a bird if the balance measured five consecutive masses of >50 g that were within a range of 5 g. These criteria were chosen to eliminate masses from birds that were perching incorrectly (e.g., by placing one foot on the food hopper), but to maximise the number of stable masses recorded from moving birds. Once a stable mass had been recorded, another stable mass could not be recorded until the balance had registered a mass <10 g, indicating that the current bird had left the perch. Balances were checked with a 100-g test mass a minimum of twice daily and calibrated if necessary. In order to control for build-up of guano on the perch over the day, balances were automatically zeroed regularly throughout the day when no bird was present on the perch. Prior to modelling the mass data, any masses greater than 120 g were removed on the grounds that such masses were at least 10 g above the maximum mass ever recorded for a starling in our laboratory and were thus likely to be the result of measurement error.

The raw masses showed a clear trend of mass increase over the day; the trajectory was typically non-linear, with mass gain generally being fastest early in the day, slowing down in the middle of the day and either peaking or speeding up again towards dusk. There is substantial random error in mass, due to the imprecision of the balances and movement of the birds whilst on the perch. Furthermore, masses were not always available at all times of every day (example data from one bird are shown in Figure 3). To estimate comparable masses for each bird, we therefore modelled how individual body mass changed as a function of time of day. As long as a minimum of 10 masses were available for a bird on a given day, the masses were fitted with a regression line (on the choice of the model see below). To remove biologically implausible outliers, any masses >10 g from the fitted line were removed and a new cubic polynomial fitted to the remaining data (Figure 3). This latter fit was used to estimate body mass at specific times of the day such as dawn or dusk. To avoid extrapolation beyond the measured data, a dawn or dusk mass was only estimated if there was a mass recording within 1 h of the estimate. Dawn was chosen as the time the lights came on (0800) and dusk as 1600, because birds often stopped foraging considerably before lights-off, reducing the number of masses available in the final hour of the day.

### 2.9. Inferential Statistics

Data were analysed using R version 3.5.1 [31]. Due to the multi-level structure of the data, with individual observations nested within birds and birds nested within aviaries, we used general linear mixed models (GLMMs) for inferential statistics. GLMMs were fitted using restricted maximum likelihood estimation (REML) in the package ‘lme4′ [32] and *p*-values were calculated using Satterthwaite’s method in the package ‘lmerTest’ [33]. We included a random intercept for bird in all GLMMs. A random intercept for aviary was also initially added to the models, but since in practice the aviary explained little or no variance, this random effect was dropped from the final version of the models reported. A fixed effect of sex was included in all models, because on average, male starlings are skeletally larger and heavier than females, meaning that there is reason to expect the effects of sex on between-subject differences in foraging behaviour and body mass. As expected, the males had longer tarsi than the females, but the difference was marginally non-significant (linear model: β_male_ ± se = 0.99 ± 0.47; F_1,10_ = 4.41, *p* = 0.062).

To assess the reliability of the measurements of reinforcements earned and body mass derived from the SFS across the 11 days of the study, we computed intra-class correlation coefficients (ICCs) and their 95% confidence intervals using the R package ‘psych’ [34]. ICCs were based on a two-way random-effects model assuming single measurements and absolute-agreement.

## 3. Results

### 3.1. Operant Training

The six previously-trained birds all started key pecking on release into their experimental aviary and began their first full day of continuous foraging the following day.

The six naïve birds took a total of eight days to learn to peck the illuminated key for food: habituation to the SFS took four days (training days 1–4) and training to key peck a further four days (training days 5–8). In the first session of unconditional auto-shaping (training day 5), four out of six birds pecked on at least one trial and in the second session (training day 6), all six birds pecked on at least three trials. In the third session (training day 7), both aviaries advanced to conditional training and all six birds pecked on at least 41 trials (Figure 4). The next day (training day 8), both aviaries advanced to continuous foraging and, since both groups were pecking well, at the end of the day, the two groups were united in a single experimental aviary and began their first full day of continuous foraging the following day.

### 3.2. Continuous Foraging

The data presented were from 11 consecutive days of continuous foraging, collected while the 12 birds were housed under a closed economy in two groups of six.

#### 3.2.1. Reinforcements Earned

Birds earned an average of 261 ± 138 reinforcements bird^−1^ × day^−1^ (mean ± sd; see Table 1 for individual descriptive statistics). The ICC for reinforcements bird^−1^·day^−1^ was 0.85 (95% CI: 0.73–0.94), indicating moderate to excellent reliability for measurement of this outcome variable. The number of reinforcements earned per day by each bird decreased slightly over the 11 days of continuous foraging (GLMM: β ± se = −3.37 ± 1.56; F_1,119_ = 4.71, *p* = 0.032); there was no significant effect of sex on number of reinforcements earned (β_male_ ± se = 120.18 ± 74.15; F_1,10_ = 2.63, *p* = 0.136). Within each day, the rate at which birds earned reinforcements declined significantly with increasing hour of the day (GLMM: β ± se = −1.50 ± 0.14; F_1,1195_ = 119.28, *p* < 0.001; Figure 5).

#### 3.2.2. Food Consumption

The dataset comprised 10 and 11 daily measurements of total food consumed from aviaries 114 and 116, respectively; day 11 from aviary 114 was missing due to experimenter error. The birds consumed 24.9 ± 1.0 g bird^−1^ × day^−1^ (mean ± se) of crumb. A GLMM with a random effect of aviary showed a non-significant increase in daily consumption over the 11 days of continuous foraging (GLMM: β ± se = 0.34 ± 0.17; F_1,18.08_ = 4.04, *p* = 0.060).

To explore whether variation in operant foraging behaviour predicted variation in food consumption, we asked whether the daily total number of reinforcements earned by all the birds in each aviary predicted the daily total crumb consumption in that aviary. A GLMM with a random effect of aviary showed that food consumption was significantly positively predicted by the number of reinforcements earned (GLMM: β ± se = 0.006 ± 0.002; F_1,18.03_ = 5.77, *p* = 0.027).

#### 3.2.3. Body Mass

The initial dataset comprised 8857 stable mass measurements, but 16 of these (0.18%) were excluded due to being greater than 120 g (see Methods), yielding 8841 masses for modelling. To establish the highest degree of polynomial necessary to model these data, we compared the fit of linear, quadratic, and cubic polynomials to the mass data from each bird day (data from the ninth day for bird P75 were not fitted due to there being too few measurements—see Methods). The best fitting model on each day was defined as the model with the lowest Akaike’s information criterion (AIC). Overall, the data from 27% of days were fitted best with a linear model, 30% with a quadratic model, and 43% with a cubic model (Figure 6). With the exception of one bird (P79), the majority of days for individual birds fitted best with either a quadratic or cubic polynomial. Therefore, there is strong support for daily mass gain being nonlinear, and specifically for the use of a cubic model capable of capturing two points of inflection. Based on these findings, we elected to use cubic polynomials to estimate how mass changed with time of day.

A further 162 masses (1.83%) were excluded as outliers during the fitting process (see Methods), meaning that the results described below were based on a total of 8679 stable masses, yielding an average of 66 ± 3 masses (mean ± sd) bird^−1^·× day^−1^. The 131 cubic polynomial fits had an R-squared value of 0.54 ± 0.20 (mean ± sd). Therefore, the cubic polynomials on average explained over half of the variation in the mass measurements. We used the fitted cubic polynomials to estimate body mass at each hour of each day between 0800 and 1600.

Selected descriptive statistics for body mass derived from this approach are shown in Table 1. The ICCs for reliability of body mass were as follows: dawn mass 0.90 (95% CI: 0.79–0.97), noon mass 0.91 (95% CI: 0.82–0.97), and dusk mass 0.89 (95% CI: 0.79–0.97). Thus, estimates of body mass had good to excellent reliability, with noon mass being the most reliable. Average noon mass was 84.1 ± 5.7 g (mean ± sd). There was no effect of day of study on noon mass (GLMM: β ± se = 0.067 ± 0.053; F_1,118_ = 1.59, *p* = 0.2100). Although, as expected for starlings, the males were heavier than females (86.9 ± 6.8 versus 81.4 ± 3.0 respectively), but the difference was not significant (β_male_ ± se = 5.44 ± 3.02; F_1,10_ = 3.24, *p* = 0.100). Mean dawn mass was highly positively correlated with mean dusk mass (Pearson correlation: r_10_ = 0.98, *p* < 0.0001).

All birds gained mass between dawn and dusk each day and the gain was 6.0 ± 1.2 g (mean ± sd). Nightly mass loss was 6.1 ± 1.2 g (mean ± sd). The ICCs for the reliability of daily mass gain and nightly loss were 0.16 (95% CI: 0.05–0.44) and 0.16 (95% CI: 0–0.45), respectively. Thus, estimates of gain and loss had poor reliability. However, mean nightly mass loss was highly positively correlated with mean daily mass gain (Pearson correlation: r_10_ = 0.99, *p* < 0.0001). Figure 7 shows how predicted mass changed with time of day for each of the 12 birds.

#### 3.2.4. Foraging Effort and Body Mass

To explore the association between foraging effort and body mass, we asked whether dusk body mass was predicted by the number of reinforcements earned during the day (equal to the number of key pecks). Dusk mass was significantly positively predicted by the number of reinforcements earned since dawn (GLMM: β ± se = 0.0086 ± 0.0030; F_1,125_ = 8.11, *p* = 0.005; Figure 8a); there was no significant effect of sex (β_male_ ± se = 3.83 ± 3.34; F_1,10_ = 1.32, *p* = 0.278).

Although we might predict that birds that are heavier overall are heavier because they eat more, this does not follow from the above result, because the mixed model combines within-subject effects due to plastic phenotypic responses, with between-subject effects due to individual differences. We therefore used the method described by van de Pol and Wright [35] to separate within- from between-subject effects of foraging effort on body mass. We used within-subject centring (i.e., subtracting the mean daily reinforcements earned by each subject from each daily observation) to derive a predictor variable that expresses the within-subject component of variation in reinforcements earned. We also derived a predictor variable that expresses the between-subject component of variation in reinforcements earned by replacing all daily observations for a given subject with the mean daily number of reinforcements earned by that subject. A second GLMM with these two new predictor variables (in place of reinforcements earned) showed that while the within-subject effect of reinforcements earned on dusk mass was significant (β ± se = 0.0098 ± 0.0031; F_1,117_ = 9.96, *p* = 0.002), the between-subject effect was not (β ± se = −0.0155 ± 0.0125; F_1,9_ = 1.54, *p* = 0.247). Furthermore, the parameter estimates for these effects were in opposite directions, with a positive association between reinforcements earned and body mass within subjects, but a negative association, albeit non-significant, between subjects (Figure 8b,c). In this model, males were marginally non-significantly heavier than females (β_male_ ± se = 6.73 ± 3.30; F_1,9_ = 4.16, *p* = 0.072).

To test whether the difference in the parameter estimates for the within- and between-subject effects of reinforcements earned was significant, we fitted a third GLMM with both the original predictor, which combines within- and between-subject effects, and our new predictor, which expresses only between-subject variation. The between-subject effect in this model represents the difference between the between- and within-subject effects in the second model [35]. This GLMM showed a significant effect of the original combined predictor (β ± se = 0.0098 ± 0.0031; F_1,117_ = 9.96, *p* = 0.002), a marginally non-significant effect of the predictor expressing between-subject variation (which in this model, tests for a difference between within- and between-subject effects; β ± se = −0.0253 ± 0.0129; F_1,10.1_ = 3.84, *p* = 0.078) and a marginally non-significant effect of sex (β_male_ ± se = 6.73 ± 3.30; F_1,9_ = 4.16, *p* = 0.072). Thus, there is some evidence for different within- and between-subject effects of the number of reinforcements earned. Whereas earning more reinforcements during the day resulted in higher dusk mass within birds, between-bird differences in dusk mass were not explained by between-bird differences in the number of reinforcements earned.

## 4. Discussion

We described a novel RFID-based technology, a social foraging system (SFS) that permits individual operant foraging behaviour and body masses of group-housed starlings to be measured from dawn until dusk, seven days a week. Our aim was to develop a refined method for studying body mass regulation in small birds that simultaneously delivered improved ecological validity, data quality, and animal welfare. We demonstrated that naïve starlings rapidly learn operant key pecking through an auto-shaping procedure while housed in groups of three birds. Once trained to forage from the SFS, starlings housed in groups of six maintained stable body masses while foraging on a ratio schedule under closed economy for a period of 11 consecutive days. Birds gained 6.0 ± 1.2 g (mean ± sd) between dawn and dusk each day and lost an equal amount overnight. Within each day, the rate of individual foraging behaviour (key pecking) decreased between dawn and dusk and birds gained mass non-linearly, with the rate of mass gain decelerating as the day progressed. There were stable individual differences in mean body mass over the 11 days of data collection. Within-bird variation in daily foraging rate positively predicted within-bird variation in dusk mass. However, between-bird variation in mean foraging rate was uncorrelated with between-bird variation in mean mass. Below, we discuss the scientific and animal welfare implications of these findings.

### 4.1. Diurnal Mass Gain Trajectories

Using data from the SFS, we were able to study how both foraging effort (key pecks) and body mass changed within the day and between days in individual birds. Measures of foraging behaviour and body mass were reliable across days within birds, but differed between birds, suggesting that the measurement error was sufficiently small to allow us to detect stable individual differences in foraging behaviour and body mass. In contrast, estimates of daily mass gain and overnight mass loss were relatively unreliable, presumably because gain and loss were computed from the difference between two measurements of mass, doubling the measurement error.

The birds foraged most intensively first thing in the morning and their rate of foraging declined as the day progressed. The majority of diurnal mass gain trajectories were fitted best with a cubic polynomial function. In the majority of birds, body mass increased most rapidly early in the day prior to levelling off later in the day. This diurnal trajectory of mass gain is predicted by optimality models when birds are constrained by only starvation risk and mass-dependent costs (i.e., predation risk) are absent or low [7]. In contrast, the most common diurnal mass gain trajectory observed in wild birds is a double-exponential function with a second period of rapid mass increase prior to dusk [36]. This latter trajectory is predicted when birds are constrained simultaneously by both starvation risk, which favours mass gain early in the day, and mass-dependent risk, which favours delaying mass gain until later in the day [7]. Interestingly, this double-exponential trajectory was also observed in caged coal tits (*Periparus ater*) caught frequently for manual weighing [37]. Over the 11 days of this study, only one starling in our (P78) showed any evidence of the double-exponential pattern in its mean mass gain trajectory, with a slight increase in the rate of mass gain prior to dusk (see Figure 7). Therefore, one interpretation of the diurnal mass trajectories that we observed was that our starlings perceived their predation risk to be low and regulated their body masses accordingly, by accumulating the majority of their daily mass gain early in the day. This interpretation could be tested by exploring whether experimentally increasing perceived predation risk (e.g., by unpredictably exposing the birds to a stuffed sparrow hawk) causes the predicted change in diurnal mass trajectories with a shift toward delaying some mass gain until later in the day.

### 4.2. Limitations on Measurement of Food Consumption

Due to constraints of our methodology, data on food consumption was only available at the aviary level and we were unable to directly measure how much food each bird consumed each day. However, we were able to demonstrate that the daily total number of reinforcements earned by all the birds in each aviary predicted the daily total food consumption in that aviary, suggesting that the number of reinforcements earned (which is available for each individual) could be used as a proxy for individual food consumption. Obtaining accurate individual data for food consumption is currently only possible in individually housed birds. Due to space constraints in most laboratories, this usually implies housing birds in smaller cages that restrict natural behaviour (and compromise animal welfare) (e.g., [37]). Therefore, there is a trade-off between the level at which some variables can be measured and the ecological validity of experiments, which has implications for the types of effects that can be detected. Given the established importance of the social and physical environment for body mass (e.g., [20,23]), we argue that a move toward more naturalistic experimental paradigms in which opportunities for social interaction and physical activity are present is critical, even if this comes at the cost of limiting individual-level measurement of some variables.

### 4.3. Food Consumption and Body Mass

Within individual birds, we observed a positive association between the foraging effort a bird made in a day (i.e., the total key pecks and hence total reinforcements earned) and its dusk mass that day. If we assume that individual differences in foraging effort relate to individual differences in food consumption, then we have evidence that within-individual variation in dusk mass is explained by the amount of food a bird consumed that day. Interestingly however, the larger between-bird differences in mean body mass that we observed were not predicted by between-bird differences in mean daily foraging effort (Figure 8c), suggesting that variation in mean body mass is not explained by the mean amount of food consumed.

In animals for which body mass is stable (as was the case in our birds), daily energy intake must equal daily energy expenditure. It follows that one possible interpretation of the lack of association between mean foraging effort and mean body mass is that there are large between-bird differences in energy expenditure that are unrelated to mean body mass. In humans, a recent analysis of variation in daily energy expenditure reported large individual differences in energy expenditure, even after controlling for effects of fat free mass, sex, and age, suggestive of large unexplained individual differences in resting metabolic rate [38]. If similar variation is proven to exist in wild-caught starlings, this species could provide a good model for research into the causes of individual variation in resting metabolic rate.

An alternative explanation for the lack of association between mean foraging effort and mean body mass is that there are large individual differences in the rate at which our birds consumed food from the hopper during the 5-s reinforcements. Further work is required to establish whether there is between-bird variation in the amount of food consumed per reinforcement earned (this could easily be measured by housing birds individually for a few days). However, even without further data, the magnitude of the individual variation seen in the number of reinforcements earned per day (see Table 1) suggests that it is likely that the birds consumed food at very different rates.

### 4.4. Animal Welfare Benefits

The SFS delivers several refinements over the individual caging typically used for operant studies in small birds [26,37,39]. The real-time data on foraging behaviour and body weight provided by the SFS allowed us to closely monitor the welfare of the starlings. Any changes in behaviour or body weight indicative of a potential welfare problem were rapidly detected and investigated. The husbandry of the birds in the SFS is also likely to have delivered welfare improvements over previous methods. The birds were group housed throughout this study, which is likely to be important for a social species such as the starling. Starlings prefer to forage in the presence of conspecifics despite their rate of food intake being slower [40]. It is likely that social species such as starlings avoid lone foraging when possible due to the increased predation risk it presents [41]. Our birds were housed in an aviary that permitted a greater range of behaviour than is possible in smaller individual cages. In addition to allowing the birds enough space to fly properly, the size of the room allowed the birds to retreat to a high perch when humans entered the aviary. Caged wild-caught starlings move to the rear of their cages when a human enters the room, indicating fear of humans [42]. It thus seems likely that the ability to escape to a high perch would reduce the stress caused by human presence. Finally, the mass data provided by the SFS allowed us to minimise the frequency with which we had to catch the birds for manual weighing. Catching and manual weighing was only required by our protocol at the beginning and end of the study and for welfare monitoring in the early stages of habituation to the SFS. In contrast, to collect comparable data on daily mass gain trajectories from coal tits, birds were caught from their cages and restrained in a box for weighing 10 times per day for a period of at least 24 consecutive days [37]. Given that catching and restraint is an established procedure for inducing acute stress in small birds [43,44,45], this latter procedure is likely to have induced chronic stress, with potential effects on body mass. As noted above, the tits in the latter study showed diurnal mass gain trajectories predicted of birds subject to mass-dependent costs, whereas our starlings did not. This difference leads to the interesting possibility that diurnal mass gain trajectories in small birds could be used as a novel welfare indicator to measure perceived predation risk.

## 5. Conclusions

The social foraging system (SFS) for European starlings simultaneously delivers improved ecological validity, data quality, and animal welfare compared with conventional methods in which birds are individually caged and caught frequently for manual weighing. The system delivers reliable estimates of individual foraging behaviour and body mass both within- and between-days. The SFS will facilitate novel research into the biology and psychology of body mass regulation including understanding the impacts of limited and unpredictable access to food (e.g., [2]).

## Figures and Tables

**Figure 1 animals-12-01159-f001:**
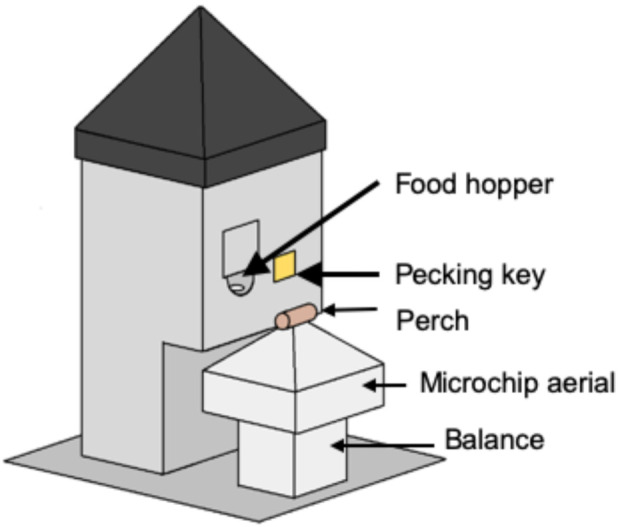
Diagram of a single social foraging system station. The wooden perch was 3.7 cm long and was mounted 22.0 cm above the base of the SFS and 8.0 cm from the hopper entrance. The pecking key was a 3.5-cm square of clear Perspex, hinged from the top and illuminated from behind with an array of LEDs. The food hopper was accessed via a 4 cm-wide aperture in the front panel of the SFS located to the left of the pecking key.

**Figure 2 animals-12-01159-f002:**
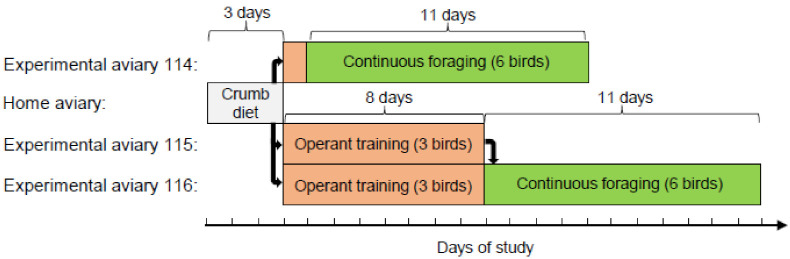
Scheme showing the study design and timeline for the 12 birds used in this study. During operant training, there was one SFS station per aviary and during continuous foraging, there were two stations per aviary. Thus, the ratio of birds per SFS station was maintained at 3:1 throughout. The entire study lasted 22 days.

**Figure 3 animals-12-01159-f003:**
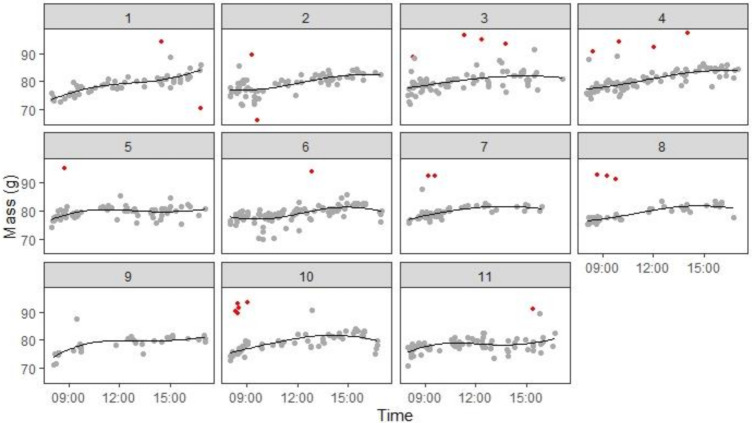
Example of raw mass data from one bird. Panels 1–11 show scatterplots of individual stable mass measurements (g) by time of day (hours) for 11 consecutive days of continuous foraging under closed economy. Red points indicate masses that were excluded from the final model due to falling more than 10 g from the initial cubic fits (see Methods for details). The black line shows the cubic fit to the remaining grey points that was used to derive the estimated masses for specific times of day. Data from bird P77 are shown; equivalent graphs for the other 11 birds are shown in Appendix A.

**Figure 4 animals-12-01159-f004:**
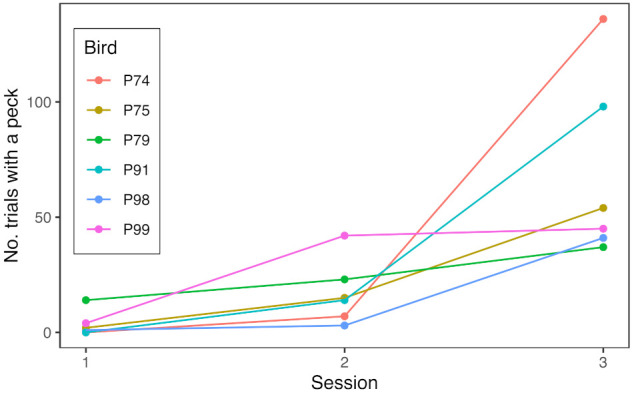
Group-housed experimentally-naïve birds rapidly acquired operant key pecking over three days of training. The six birds were housed in two single-sex groups of three during training (male group: P74, P79, and P98; female group: P75, P91, and P99). Sessions 1–3 took place on training days 5–7, following habituation to the SFS. Line graphs show the number of trials on which each bird pecked the lit key on the SFS by session. In sessions 1 and 2 (auto-shaping; 90 trials each), access to food was unconditional on key pecking, whereas in session 3 (270 trials), access to food was conditional on a bird pecking the lit key.

**Figure 5 animals-12-01159-f005:**
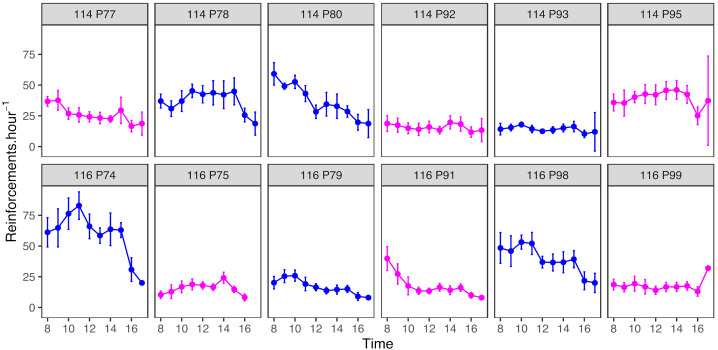
Rate of earning reinforcements declines over the day. Panels show scatterplots of the average reinforcement rate (mean ± 95% CI) by time of day for each of the 12 birds. Each row of panels corresponds to one experimental aviary. Data from male birds are shown in blue and females in pink.

**Figure 6 animals-12-01159-f006:**
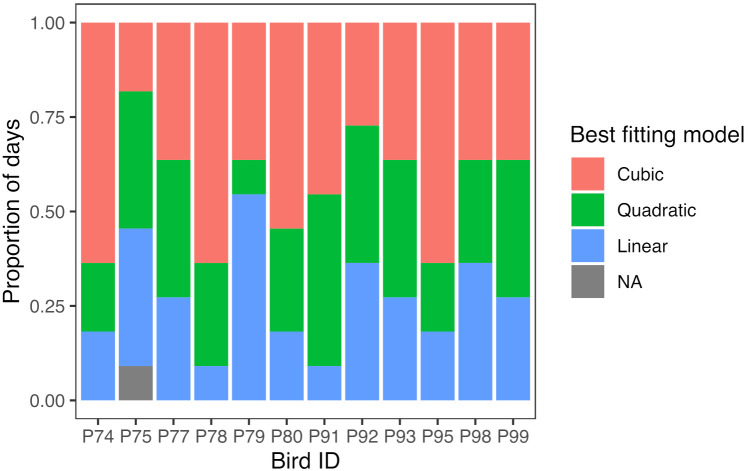
Best-fitting polynomial models describing the association between time of day and body mass. Stacked bar chart showing the proportion of days for each bird that were fitted best by a linear, quadratic, or cubic polynomial model. The ninth day of data for bird P75 was not fitted due to insufficient data.

**Figure 7 animals-12-01159-f007:**
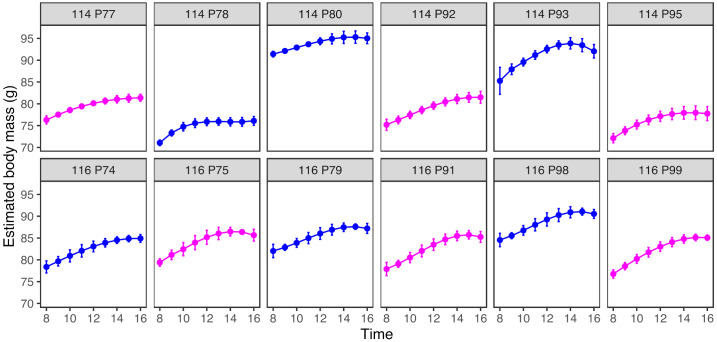
Body mass increased non-linearly over the course of the day, slowing or peaking toward dusk. Panels show the mean predicted mass (±95% CI) by time of day for each of the 12 birds. Each row of panels corresponds to one experimental aviary. Data from male birds are shown in blue and females in pink.

**Figure 8 animals-12-01159-f008:**
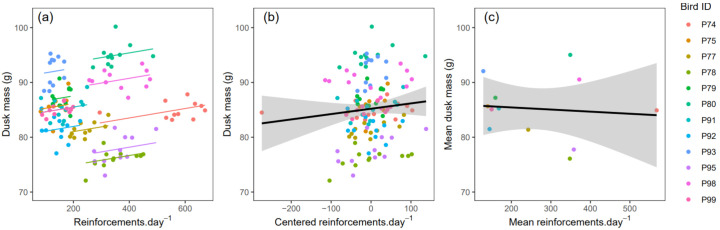
The positive effect of foraging effort (daily reinforcements earned) on dusk body mass is driven by a within-subjects effect. (**a**) Scatterplot of dusk mass by the total number of reinforcements·× day^−1^ with each point representing the data from one bird day. Lines show the fitted values from a standard mixed model that combines within- and between subject effects. (**b**) Scatterplot of dusk mass by within-subject centred reinforcements·× day^−1^ with each point representing the data from one bird on one day. The line shows the significant positive within-subject effect. (**c**) Scatterplot of mean dusk mass by mean reinforcements × day^−1^ with points representing birds. The line shows the absence of a positive between-subject effect.

**Table 1 animals-12-01159-t001:** Details of birds and descriptive statistics (mean ± sd) for reinforcements earned and body mass data ^†^.

Aviary	Bird ID	Sex	Tarsus Length (mm)	Reinforcements × Day^−1^	Masses·Day^−1^	Dawn × Mass	Noon × Mass	Dusk × Mass	Daily × Gain	Nightly × Loss
114	P77	F	28.8	243 ± 47	71 ± 20	76.3 ± 1.6	80.1 ± 0.9	81.4 ± 1.4	5.1 ± 1.8	5.0 ± 1.7
114	P92	F	30.5	145 ± 44	44 ± 17	75.2 ± 2.2	79.6 ± 1.4	81.5 ± 2.3	6.3 ± 3.0	6.4 ± 3.9
114	P95	F	29.0	358 ± 65	127 ± 40	72.1 ± 1.8	77.1 ± 2.0	77.8 ± 2.7	5.6 ± 2.9	6.0 ± 1.9
114	P78	M	29.4	348 ± 68	41 ± 10	71.1 ± 1.1	75.9 ± 1.5	76.1 ± 1.7	5.0 ± 1.9	4.9 ± 2.3
114	P80	M	29.9	349 ± 56	69 ± 14	91.4 ± 1.1	94.4 ± 1.3	95.0 ± 2.1	3.6 ± 2.2	3.8 ± 2.0
114	P93	M	30.5	129 ± 23	25 ± 11	85.3 ± 5.2	92.6 ± 1.4	92.1 ± 2.6	6.9 ± 5.8	7.1 ± 6.7
116	P75	F	29.6	140 ± 33	31 ± 11	79.4 ± 1.5	85.2 ± 2.7	85.7 ± 2.3	6.5 ± 2.7	6.3 ± 3.1
116	P91	F	30.5	168 ± 51	81 ± 30	77.9 ± 2.6	83.5 ± 2.3	85.3 ± 2.1	7.4 ± 3.1	7.3 ± 2.9
116	P99	F	28.4	150 ± 38	63 ± 27	76.8 ± 1.6	83.0 ± 1.9	85.1 ± 1.0	8.3 ± 1.5	8.3 ± 1.6
116	P74	M	30.6	568 ± 103	107 ± 38	78.4 ± 2.3	83.1 ± 2.1	84.9 ± 1.5	6.5 ± 2.5	6.4 ± 2.6
116	P79	M	30.9	160 ± 26	42 ± 15	82.0 ± 2.6	86.0 ± 2.3	87.2 ± 2.0	5.2 ± 3.6	5.2 ± 3.0
116	P98	M	31.5	372 ± 81	91 ± 38	84.5 ± 2.6	89.3 ± 2.6	90.5 ± 1.7	6.0 ± 1.5	6.2 ± 2.7

^†^ Mass statistics are based on fits from 11 days of continuous foraging for all birds except P75, which had 10 days due to insufficient data for fitting on one day. ‘Masses × day^−1^′ is the number of stable masses remaining after all exclusions (see Methods for details). ‘Dawn mass’ is the fitted mass at 0800 h, ‘Noon mass’ is the fitted mass at 1200 h, and ‘Dusk mass’ is the fitted mass at 1600 h. ‘Daily gain’ is the difference between dawn mass and dusk mass. ‘Nightly loss’ is the difference between the dusk mass the previous day and dawn mass. All masses are in grams.

## Data Availability

Data and the analysis script are available at: https://doi.org/10.5281/zenodo.6368487, published on 18 March 2022.

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
