# Peer review of "A Refined Method for Studying Foraging Behaviour and Body Mass in Group-Housed European Starlings"

_animals, 2022, doi:10.3390/ani12091159_

Round 1

Reviewer 1 Report

I enjoyed reading this paper, which is very clearly written and suggests an original approach to studying foraging in birds, especially in the case of species foraging in flocks. The Social Foraging System also represents a semi-natural environment that requires minimal human intervention to collect relevant behavioral data. I only have minor comments that I would like to be responded and then recommend the publication of this work.

Lines 75-76: “reduced energy expenditure is potentially the more important mechanism, with some evidence for reductions in physical activity”. In the context of food unpredictability (discussed just above), can we really say this? Why should captive birds exposed to unpredictable vs. predictable food conditions experience different levels of physical activity, if tested in the same apparatus? I am not necessarily suggesting that a difference in food consumption is responsible for a difference in body mass, but I am not sure if I understand the argument about physical activity as a major explanatory factor in this context.

Lines 118-120: “subjects of social species should be housed in groups that permit social interactions and the formation of dominance hierarchies”. This is a traditionally neglected factor, which should be taken into account to guarantee the ecological validity of studies involving social foragers. But dominants and subordinates often show differences in body mass. This could potentially be a problem in case researchers want to test the influence of other factors than dominance on body mass. Could you elaborate a bit more on this, maybe in the discussion?

Lines 257-258: “dried mealworms were placed in the SFS food hopper entrance to attract the birds”. Was the food manually placed? Was the procedure repeated several times a day? Could all birds have access to these food items? This should be specified, as this learning step is crucial for the rest of the experiment.

Lines 410-411: “the two groups were united in a single experimental aviary”. Was this aviary a new one, or one of those in which 3 starlings had previously been trained? In this latter case, should we not expect problems of territorial dominance and more difficult access to the hoppers among the newly introduced individuals?

Lines 624-625: “suggesting that variation in mean body mass is not explained by the mean amount of food consumed”. Not sure if I fully agree with this conclusion. You showed that dusk mass positively correlates with foraging effort within individual birds, and this is for me a major point. The fact that this finding is not reproduced after considering the larger between-bird differences in mean dusk mass may just indicate that other factors (than food consumption) influence body mass. For example, the difference in body mass between obese and leaner people does not necessarily reflect a difference in the amounts of food ingested. But one specific individual has a higher probability of becoming obese in a consumerist society. I think it would be interesting to know whether, in your study, the individuals with the largest differences in body mass at dusk are also those with the largest differences at dawn. If yes, this would indeed confirm that the observed difference at dusk can be attributed to factors other than consumption—such as differences in metabolic rates, etc.

Author Response

Lines 75-76: “reduced energy expenditure is potentially the more important mechanism, with some evidence for reductions in physical activity”. In the context of food unpredictability (discussed just above), can we really say this? Why should captive birds exposed to unpredictable vs. predictable food conditions experience different levels of physical activity, if tested in the same apparatus? I am not necessarily suggesting that a difference in food consumption is responsible for a difference in body mass, but I am not sure if I understand the argument about physical activity as a major explanatory factor in this context.

We were simply trying to make the point that energy balance depends on both energy intake and energy expenditure and that the latter could be affected by the environment in which an animal is kept. We have reworded this sentence as follows: ‘While it is generally assumed that increased body mass under conditions of unpredictable food results from increased energy intake [16–18], reduced energy expenditure is also likely to be important; studies of both starlings and zebra finches (Taeniopygia guttata) show that birds respond to unpredictable food by reducing their physical activity [2,19].

Lines 118-120: “subjects of social species should be housed in groups that permit social interactions and the formation of dominance hierarchies”. This is a traditionally neglected factor, which should be taken into account to guarantee the ecological validity of studies involving social foragers. But dominants and subordinates often show differences in body mass. This could potentially be a problem in case researchers want to test the influence of other factors than dominance on body mass. Could you elaborate a bit more on this, maybe in the discussion?

We acknowledge this point and pointed out that dominants and subordinates have different body masses earlier in the introduction at lines 91-100.

Lines 257-258: “dried mealworms were placed in the SFS food hopper entrance to attract the birds”. Was the food manually placed? Was the procedure repeated several times a day? Could all birds have access to these food items? This should be specified, as this learning step is crucial for the rest of the experiment.

The dried mealworms were manually placed in the hopper entrance to attract the birds to the hopper. We have added this detail to the text. We don’t know whether all the birds in an aviary accessed the dried mealworms. It is possible that only one or two birds ate the worms, but that this served to attract the others to the hopper through a social learning mechanism (local enhancement). However, as stated in the text, we were able to monitor feeding from the hopper via CCTV and did not progress to the next stage of training until all the birds were definitely feeding from the hopper.

Lines 410-411: “the two groups were united in a single experimental aviary”. Was this aviary a new one, or one of those in which 3 starlings had previously been trained? In this latter case, should we not expect problems of territorial dominance and more difficult access to the hoppers among the newly introduced individuals?

The three birds trained in aviary 115 were united with the three birds trained in aviary 116 in aviary 116, as indicated in Figure 2. Starlings are not territorial and prefer to forage in groups. We did not notice any reluctance of the newly-introduced birds to forage from the SFS stations in the new aviary. Note that when the 6 birds were united in aviary 116 a second SFS station was provided in 116 to maintain the ratio of birds to SFS stations at 3:1. This detail has been added to the caption to the caption of Figure 2: “During operant training there was one SFS station per aviary and during continuous foraging there were two stations per aviary. Thus, the ratio of birds per SFS station was maintained at 3:1 throughout.”

Lines 624-625: “suggesting that variation in mean body mass is not explained by the mean amount of food consumed”. Not sure if I fully agree with this conclusion. You showed that dusk mass positively correlates with foraging effort within individual birds, and this is for me a major point. The fact that this finding is not reproduced after considering the larger between-bird differences in mean dusk mass may just indicate that other factors (than food consumption) influence body mass. For example, the difference in body mass between obese and leaner people does not necessarily reflect a difference in the amounts of food ingested. But one specific individual has a higher probability of becoming obese in a consumerist society. I think it would be interesting to know whether, in your study, the individuals with the largest differences in body mass at dusk are also those with the largest differences at dawn. If yes, this would indeed confirm that the observed difference at dusk can be attributed to factors other than consumption—such as differences in metabolic rates, etc.

I don’t think there is any disagreement here; we are arguing exactly what this reviewer suggests. The mean dawn mass of a bird is highly correlated with the mean dusk mass of a bird in our dataset (0.98). We have added this analysis to the results: “Mean dawn mass was highly positively correlated with mean dusk mass (Pearson correlation: r10 = 0.99, p < 0.0001).” Thus, the individual differences in dusk mass reflect stable individual differences in body weight. We have added the following sentence to the first paragraph of the discussion: “There were stable individual differences in mean body mass over the 11 days of data collection.” We have also removed the reference to dusk mass from the critical sentence highlighted above, which I hope makes our meaning clearer: “Interestingly however, the larger between-bird differences in mean mass that we observed were not predicted by between-bird differences in mean daily foraging effort (Figure 8c), suggesting that variation in mean body mass is not explained by the mean amount of food consumed.”

Reviewer 2 Report

I enjoyed reading this manuscript very much.

I have only two minor suggestions. Please include (1) the size and shape of the aviary and (2) further specifications of the new social foraging system station (e.g., length, height, and depth in figure 1).

Author Response

I have only two minor suggestions. Please include (1) the size and shape of the aviary and (2) further specifications of the new social foraging system station (e.g., length, height, and depth in figure 1).

The dimensions of the aviaries are clearly stated in the methods:” (280 cm wide × 300 cm deep × 255 cm high).” As stated, all were identical. We have added the following information about the important dimensions of the SFS to the legend of Figure 1: “The wooden perch was 3.7 cm long and was mounted 22.0 cm above the base of the SFS and 8.0 cm from the hopper entrance. The pecking key was a 3.5-cm square of clear Perspex, hinged from the top and illuminated from behind with an array of leds. The food hopper was accessed via a 4 cm-wide aperture in the front panel of the SFS located to the left of the pecking key.”

Reviewer 3 Report

Birds face several constraints in regulating body mass throughout the day. Two constraints stand out: early recovery of fat reserves depleted at night and avoiding getting fat too early due to increased flight costs and predation risk. These are classic research topics, as clearly explained in the manuscript. And despite the accumulated research effort to improve the realism of experiments or to understand the effects of variability and unpredictability in foraging (e.g., Kacelnik and Bateson 1996; Bateson and Kacelnik 1998; Kacelnik and El Mouden 2013), the daily regulation of body mass in birds has failed to be fully explained by a general theory. Improved experimental designs are required. The one developed by Bateson and Nolan fulfils this requirement.

The paper is written with clarity and logically structured. A few minor questions must be addressed. Admittedly, some of them could highlight my misunderstanding, so just answer them in the letter to the editor.

Minor comments

Lines 177-178. The lights were on between 0800 and 1715. The lights dimmed gradually at 1700 to simulate dusk. Perhaps they were also gradually up between 0800 and 0815, but no explanation is provided in the text. Were they? Perhaps the dawn was not gradual but sudden. Please, explain how the experimental set up provided a smooth dawn. 
Perhaps the birds were aware of the time of day if they perceived visual or auditory stimuli outside the aviaries. Since the temperature was stable (18-20 C, line 176), it can be inferred that the aviaries were isolated from the outside. Under these conditions, the air in the room must be renewed according to the animals' number, size, and metabolism. Perhaps the ARRIVE 2.0 guidelines provide the amount of external air (L) per unit time, per aviary size (m^3), per subject and their BMR (basal metabolic rate). Rooms with open windows allow birds to breathe but also to be aware of the time of day. They also prevent the control of air temperature. I assume that the aviary was isolated from the outside and therefore had an automatic system to renew the air with internal and external air mixture. Please provide the rate of air renewal (per hour and per bird, for example) in one sentence.

Line 342. The gross masses included in Figure 1 and Supplementary Figures S1-S11 do not show a trend of increase throughout the day because the scale of the vertical axis is too wide. For example, Supplementary Figure S1 sets the weight scale from 50 to 100 grams, but starling P74 increased its weight between dawn and dusk by only 6.5 grams (Table 2). As a result, there is no discernible trend of increase in the black lines, which appear almost horizontal on a 50-gram scale. To maintain the wording of line 342, I suggest redrawing the Figures without the red outliers and the scale adjusted to the minimum and maximum gross masses included in the statistical analyses. The current Figures can be kept in the supplementary material because they are helpful to show the outliers, but as additional figures to the new ones

Line 356. Dusk was 1600, but on line 178, dusk was at 1715. Between 1600 and 1715, the birds stopped foraging considerably before the lights went out. This line may hold the key to understanding the results. The animals could obtain the food needed to survive the night in a fraction of the day. Perhaps it might take a few more days in the aviaries for the animals to distribute their food between the beginning and end of the day. A glance at the eleventh day suggests that the trajectories of some individuals began to form the typical double exponential described in the literature. Although eleven days may have been sufficient for gross weight to be stable between days, trajectories in daily weight gain could change over more extended periods. This is a possibility that should be explored. Most studies of captive birds must conform to a time limit, sometimes imposed by ethical committees. A short period of (for example) 11 days may prevent the double exponential trajectory from being achieved. The plasticity of trajectory change is inversely associated with weight or size. Smaller birds may require fewer days of experimentation and starlings more than eleven. Please, consider this possibility to enrich the Discussion.

Line 442. The data presented are from the first 11 consecutive days of continuous foraging. From the first or from the last? Maybe both, because birds were 11 days in the continuous foraging treatment. The word 'last' is confusing because it suggests birds could be in that treatment for more days.

Lines 446-448. The manuscript conveys the impression that statistical results are marginally non-significant when p-values are between 0.10 and 0.05. However, in line 382, the tarsi are described as longer in males than females, based upon a p-value of 0.062. According to the prevailing style of the manuscript, the sexual difference was (marginally) non-significant. Please, check the statistics and ensure p-values between 0.10 and 0.05 does not provide misleading support to non-significant statistical differences. I am aware this is not a big issue, so be indulgent with this comment. There are recent advocates of improving the wording of sentences when presenting statistical results on the (fuzzy?) border of significance (Muff et al. 2022a; b).

Line 491. Typo: Perhaps ', but' must be deleted. A sentence must be added otherwise.

Line 561. Birds gained 6.0 g (mean). This mean was calculated with 12 birds. I would suggest adding at the bottom of Table 2 a summary line with means (N=12 birds) of all columns. The reader needs to read the main text to find the means in the current manuscript.

References

Bateson M, Kacelnik A (1998) Risk-sensitive foraging: decision making in variable environments. p. 297-341. In Dukas R (ed.), Cognitive ecology: the evolutionary ecology of information processing and decision making. The University of Chicago Press, Chicago.

Kacelnik A, Bateson M (1996) Risky Theories: The Effects of Variance on Foraging Decisions. Integrative and Comparative Biology 36(4):402-434. https://doi.org/https://doi.org/10.1093/icb/36.4.402

Kacelnik A, El Mouden C (2013) Triumphs and trials of the risk paradigm. Anim Behav 86(6):1117-1129. https://doi.org/10.1016/j.anbehav.2013.09.034

Muff S, Nilsen EB, O'Hara RB, Nater CR (2022a) Response to 'Why P values are not measures of evidence' by D. Lakens. Trends Ecol Evol 37(4):291-292. https://doi.org/10.1016/j.tree.2022.01.001

Muff S, Nilsen EB, O'Hara RB, Nater CR (2022b) Rewriting results sections in the language of evidence. Trends Ecol Evol 37(3):203-210. https://doi.org/10.1016/j.tree.2021.10.009

Author Response

Lines 177-178. The lights were on between 0800 and 1715. The lights dimmed gradually at 1700 to simulate dusk. Perhaps they were also gradually up between 0800 and 0815, but no explanation is provided in the text. Were they? Perhaps the dawn was not gradual but sudden. Please, explain how the experimental set up provided a smooth dawn. 
Perhaps the birds were aware of the time of day if they perceived visual or auditory stimuli outside the aviaries. Since the temperature was stable (18-20 C, line 176), it can be inferred that the aviaries were isolated from the outside. Under these conditions, the air in the room must be renewed according to the animals' number, size, and metabolism. Perhaps the ARRIVE 2.0 guidelines provide the amount of external air (L) per unit time, per aviary size (m^3), per subject and their BMR (basal metabolic rate). Rooms with open windows allow birds to breathe but also to be aware of the time of day. They also prevent the control of air temperature. I assume that the aviary was isolated from the outside and therefore had an automatic system to renew the air with internal and external air mixture. Please provide the rate of air renewal (per hour and per bird, for example) in one sentence.

The aviaries that we used are licensed by the UK Home Office for housing birds. We have added a sentence to further explain the conditions in the aviary: “The aviary was without windows and was artificially lit and ventilated.”

Line 342. The gross masses included in Figure 1 and Supplementary Figures S1-S11 do not show a trend of increase throughout the day because the scale of the vertical axis is too wide. For example, Supplementary Figure S1 sets the weight scale from 50 to 100 grams, but starling P74 increased its weight between dawn and dusk by only 6.5 grams (Table 2). As a result, there is no discernible trend of increase in the black lines, which appear almost horizontal on a 50-gram scale. To maintain the wording of line 342, I suggest redrawing the Figures without the red outliers and the scale adjusted to the minimum and maximum gross masses included in the statistical analyses. The current Figures can be kept in the supplementary material because they are helpful to show the outliers, but as additional figures to the new ones

We have decided not to implement this suggestion. We prefer to retain Figure 3 with the excluded outliers shown for full transparency. The point of Figure 3 was to show what the raw mass data look like and this required an aspect ratio to the graphs that allowed the data points to be spread out horizontally. The increase in mass over the day for each bird is clearly shown in Figure 7.

Line 356. Dusk was 1600, but on line 178, dusk was at 1715. Between 1600 and 1715, the birds stopped foraging considerably before the lights went out. This line may hold the key to understanding the results. The animals could obtain the food needed to survive the night in a fraction of the day. Perhaps it might take a few more days in the aviaries for the animals to distribute their food between the beginning and end of the day. A glance at the eleventh day suggests that the trajectories of some individuals began to form the typical double exponential described in the literature. Although eleven days may have been sufficient for gross weight to be stable between days, trajectories in daily weight gain could change over more extended periods. This is a possibility that should be explored. Most studies of captive birds must conform to a time limit, sometimes imposed by ethical committees. A short period of (for example) 11 days may prevent the double exponential trajectory from being achieved. The plasticity of trajectory change is inversely associated with weight or size. Smaller birds may require fewer days of experimentation and starlings more than eleven. Please, consider this possibility to enrich the Discussion.

We fully agree that the shape of the birds’ daily mass gain trajectories needs further exploration and this is exactly what is facilitated by the SFS and suggested in lines 606-613 of our discussion. We have edited the relevant paragraph of the discussion to emphasise the fact that our data were only based on 11 days of data (line 604), but have chosen not to speculate further on the stability of the trajectories we report because we don’t have anything useful to say about this at present.

Line 442. The data presented are from the first 11 consecutive days of continuous foraging. From the first or from the last? Maybe both, because birds were 11 days in the continuous foraging treatment. The word 'last' is confusing because it suggests birds could be in that treatment for more days.

We have deleted the word “first” from this sentence to remove any confusion.

Lines 446-448. The manuscript conveys the impression that statistical results are marginally non-significant when p-values are between 0.10 and 0.05. However, in line 382, the tarsi are described as longer in males than females, based upon a p-value of 0.062. According to the prevailing style of the manuscript, the sexual difference was (marginally) non-significant. Please, check the statistics and ensure p-values between 0.10 and 0.05 does not provide misleading support to non-significant statistical differences. I am aware this is not a big issue, so be indulgent with this comment. There are recent advocates of improving the wording of sentences when presenting statistical results on the (fuzzy?) border of significance (Muff et al. 2022a; b).

We have now consistently referred to p-values between 0.05 and 0.1 as marginally non-significant.

Line 491. Typo: Perhaps ', but' must be deleted. A sentence must be added otherwise.

We have deleted the stray “but” from the end of this sentence.

Line 561. Birds gained 6.0 g (mean). This mean was calculated with 12 birds. I would suggest adding at the bottom of Table 2 a summary line with means (N=12 birds) of all columns. The reader needs to read the main text to find the means in the current manuscript.

We debated adding this row to Table 2 but decided not to do this, because the overall statistics are reported in the text and we would therefore be unnecessarily duplicating information. We make the full dataset available, meaning that anyone wanting alternative statistics to the ones we report is free to calculate these.